



# Viscosity and physical state of sucrose mixed with ammonium sulfate droplets

Rani Jeong[1], Joseph Lilek[2], Andreas Zuend[2], Rongshuang Xu[3], Man Nin Chan[3,4], Dohyun Kim[1], Hi Gyu Moon[5], Mijung Song[1,6*]

[1]Department of Environment and Energy, Jeonbuk National University, Republic of Korea
[2]Department of Atmospheric and Oceanic Sciences, McGill University, Montréal, Quebec, Canada
[3]Earth System Science Programme, Faculty of Science, The Chinese University of Hong Kong, Hong Kong, Hong Kong SAR, China
[4]The Institute of Environment, Energy and Sustainability, The Chinese University of Hong Kong, Hong Kong, Hong Kong SAR, China
[5]Center for Ecological Risk Assessment, Korea Institute of Toxicology (KIT), Jinju 52834, Republic of Korea
[6]Department of Earth and Environmental Sciences, Jeonbuk National University, Republic of Korea

Correspondence to: Mijung Song (mijung.song@jbnu.ac.kr)

**Abstract.** Although knowledge of the physical state of aerosol particles is essential to understand atmospheric chemistry model and measurements, information on the viscosity and physical state of aerosol particles consisting of organic and inorganic salts are still rare. Herein, we quantified viscosities at $293 \pm 1$ K upon dehydration for the binary systems, sucrose/$H_2O$ and ammonium sulfate (AS)/$H_2O$, and the ternary systems, sucrose/AS/$H_2O$ for organic-to-inorganic dry mass ratios (OIRs) = 4:1, 1:1, and 1:4. For binary systems, the viscosity of sucrose/$H_2O$ particles gradually increased from $\sim 6 \times 10^{-1}$ to $> \sim 1 \times 10^8$ Pa·s when the relative humidity (RH) decreased from $\sim 83\%$ to $\sim 24\%$, which agrees with previous studies. The viscosity of AS/$H_2O$ particles remained in the liquid state ($< 10^2$ Pa·s) for RH $> \sim 50\%$, and for RH $\leq \sim 50\%$, the particles showed viscosity of $> \sim 1 \times 10^{12}$ Pa·s, corresponding to a solid state. For ternary systems, the viscosity of organic-rich particles (OIR = 4:1) gradually increased from $\sim 4 \times 10^{-2}$ to $\sim 1 \times 10^8$ Pa·s for a RH decrease from $\sim 80\%$ to $\sim 18\%$, similar to the sucrose/$H_2O$ particles. In the particles for OIR = 1:1, the viscosities ranged smaller than $\sim 4 \times 10^1$ for RH $> 34\%$, and $> \sim 1 \times 10^8$ Pa·s at $\sim 27\%$ RH. Compared to the organic-rich particles, in the inorganic-rich particles (OIR = 1:4), drastic enhancement in viscosity was observed as RH decreased; the viscosity enhanced approximately 8 orders of magnitude in RH from 43% to 25%. Based on viscosity data, all particles studied in this work were observed to exist as a liquid, semi-solid or solid depending on the RH. Furthermore, we compared the measured viscosities of ternary systems with OIRs of 4:1, 1:1, and 1:4 to the predicted viscosities using the Aerosol Inorganic–Organic Mixtures Functional groups Activity Coefficients Viscosity model (AIOMFAC-VISC) predictions with the Zdanovskii–Stokes–Robinson (ZSR)-style organic–inorganic mixing model, with excellent model–measurement agreement for all OIRs.



## 1 Introduction

Aerosol particles can play an important role in air quality, climate change, and human health (Kulmala et al., 2011; Zhang et al., 2015; IPCC 2019; Bhattarai et al., 2020). These aerosol particles comprise mainly organic materials and inorganic salts which are frequently internally mixed in particles found in the atmosphere (Murphy et al., 2006; Zhang et al., 2007; Jimenez et al., 2009; Wang et al., 2016). Depending on the chemical composition, air temperature and relative humidity (RH), aerosol particles can undergo liquid–solid and liquid–liquid phase transitions (Martin, 2000; Marcolli, 2014) leading to different physical states. Although such data are useful to inform and validate process models and parameterizations, measurements of various physical states and changes in phase viscosities of mixed organic-inorganic aerosols still remain an understudied area. Physical states (i.e. liquid, semi-solid, and solid) of aerosol particles or phases thereof can be determined from their dynamic viscosities; a viscosity of less than $10^2$ Pa·s indicates a liquid state, a viscosity between $10^2$ and $10^{12}$ Pa·s indicates a semi-solid state, and a viscosity of greater than $10^{12}$ Pa·s indicates a solid state (Zobrist et al., 2008; Koop et al., 2011; Kulmala et al., 2011). Most studies of viscosities and the related physical states of aerosol particles have been carried out by means of laboratory studies, because there are almost no direct instruments to quantify the viscosities of ambient, airborne submicron- and supermicron-sized aerosol particles. Numerous laboratory studies on viscosities and physical states undertaken so far have focused on organic aerosols with different types of secondary organic aerosols investigated (Virtanen et al., 2010; Kuwata and Martin, 2012; Perraud et al., 2012; Renbaum-Wolff et al., 2013a; O'brien et al., 2014; Bateman et al., 2015; Song et al., 2015; Athanasiadis et al., 2016; Grayson et al., 2016; Hosny et al., 2016; Song et al., 2016a; Yli-Juuti et al., 2017; Ham et al., 2019; Petters et al., 2019; Song et al., 2019; Gervasi et al., 2020; Song et al., 2021). These studies showed that the viscosities of organic aerosol particles can vary depending on RH and chemical composition leading to different physical states, including amorphous solid (glassy), (semi-)solid, as well as liquid-like phases.

Compared to the number of studies on the viscosity of organic aerosol particles, only a few studies exist for organic compounds mixed with inorganic salts. From laboratory studies, Power et al. (2013) and Rovelli et al. (2019) observed a decrease in viscosity in mixed sucrose/inorganic salt/$H_2O$ particles at a certain RH with an increase in the inorganic fraction of NaCl and $NaNO_3$. Richards et al. (2020b) showed that higher inorganic fraction enhanced organic-inorganic aerosol viscosity at a certain RH due to the ion–molecule interaction. Song et al. (2021) showed that sucrose/$Mg(NO_3)_2$/$H_2O$ and sucrose/$Ca(NO_3)_2$/$H_2O$ particles were present in liquid to semi-solid states or even in solid state over the different RH ranges (when probing from high to low RH). From field measurements, Bateman et al. (2017) reported that atmospheric aerosol particles containing organic materials and inorganic salts in central Amazonia are in a nonliquid state with a viscosity greater than $10^2$ Pa·s even at ~95% RH during pollution events. On the other hand, Liu et al. (2019) observed that the phase of ambient aerosol particles in an urban area of China to be in a liquid state at RH greater than 60%, when the high nitrate fraction was monitored. For a more comprehensive understanding of particle phase viscosities and the related physical states of aerosol particles consisting of a mixture of organic materials and inorganic salts, more datasets are required.





Information of physical states of aerosol particles is critical to understanding heterogeneous reactions between gaseous species and aerosol particles, including the time scales and distinction of possible surface and bulk reactions. Studies have shown that the uptake coefficient of oxidants in the gas phase varied significantly depending on the physical states of involved aerosol particles (Xiao et al., 2011; Slade and Knopf, 2014; Davies and Wilson, 2015; Steimer et al., 2015; Berkemeier et al., 2016; Xu et al., 2020; Lam et al., 2021). For example, Steimer et al. (2015) showed that the ozone uptake coefficient of semi-solid particles was approximately one order of magnitude less than that of liquid particles.

To get more insights into the viscosities, and related physical states of aerosol particles, we first quantified viscosities of binary mixtures of sucrose/$H_2O$ and ammonium sulfate (AS)/$H_2O$ particles, and also ternary mixtures of sucrose/AS/$H_2O$ for organic-to-inorganic dry mass ratios (OIRs) = 4:1, 1:1, and 1:4 under dehydration conditions at $293 \pm 1$ K. Sucrose was chosen as the model organic substance because it has been investigated as a model surrogate species for secondary organic aerosols (SOA) in previous studies (Grayson et al., 2016; Song et al., 2016b; Rovelli et al., 2019; Song et al., 2021). Moreover, sucrose is a good choice for laboratory studies, because it does not easily crystallize while aqueous particles are dried, despite it not being as compositionally or functionally complex as the mixtures of hundreds to thousands of organic compounds likely comprising ambient SOA. AS was used as the model inorganic salt because it has well-defined thermodynamic properties (Braban and Abbatt, 2004; Yeung et al., 2009), and is one of most abundant species in the atmosphere (Zhang et al., 2007; Jimenez et al., 2009). Next, we determined the physical states of the particles as a function of RH based on the viscosity of the binary and ternary mixtures. Finally, the measured viscosities for sucrose/AS/$H_2O$ droplets with OIRs of 4:1, 1:1, and 1:4 are compared to predictions by the Aerosol Inorganic–Organic Mixtures Functional groups Activity Coefficients Viscosity model (AIOMFAC-VISC) when employing a Zdanovskii–Stokes–Robinson (ZSR)-style mixing model for viscosity contributions in aqueous organic–inorganic mixtures.

## 2 Experimental

### 2.1 Preparation of particles

Sucrose (99.5% purity, Sigma-Aldrich) and AS (99.9% purity, Sigma-Aldrich) were dissolved in high purity water (18 MΩ cm, Merck Millipore Synergy, Germany) to make binary and ternary aqueous mixtures with ~10 wt.% solute concentration. Selected OIRs of 4:1 (organic-rich), 1:1, and 1:4 (inorganic-rich) were used for ternary systems. In all experiments, to generate droplets, the solution was sprayed onto a substrate with a hydrophobic coating (Knopf, 2003).

### 2.2 Determination of viscosity using bead-mobility and poke-and-flow techniques

For viscosity experiments, the substrate containing droplets was placed in a RH-controlled flow-cell coupled to an optical microscope (Olympus CKX53 with 40× objective, Japan) (Pant et al., 2006; Ham et al., 2019). During the experiments, RH was continuously monitored using a digital humidity sensor (Sensirion, SHT C3, Switzerland). The uncertainty in the RH was $\pm 1.5\%$, which was calibrated by observing the deliquescence RH of $K_2CO_3$ (44% RH), NaCl (76% RH), and $(NH_4)_2SO_4$ (80%





95 RH) at 293 ± 1 K (Winston and Bates, 1960). At the start of the bead-mobility and poke-and-flow experiments, the droplets on a hydrophobic substrate were equilibrated at ~90% for ~20 min. Subsequently, the RH was decreased to target-RH with an equilibrium time of ~30 min for bead-mobility and > ~2 h for poke-and-flow experiments based on the conditioning time of sucrose/$H_2O$ (Grayson et al., 2015). During the experiments, optical morphologies of the particles, 20 – 100 μm in diameter, were monitored and recorded. All viscosity-experiments were conducted at 293 ± 1 K.

100 The bead-mobility technique has been used to quantify viscosities of aerosol particles in the range of $10^{-3}$ to $10^3$ Pa·s (Renbaum-Wolff et al., 2013b; Song et al., 2015; Song et al., 2021). The detailed procedure involved in the bead-mobility technique has been described by previous studies (Renbaum-Wolff et al., 2013a). To outline it briefly, insoluble melamine beads of ~1 μm size (Cat. no. 86296, Sigma-Aldrich) dispersed in pure water were nebulized onto the droplets, which are deposited on a hydrophobic substrate, and the substrate was then mounted in a flow-cell as described above. With a total gas flow of ~1200

105 sccm at target RH at 293 ± 1 K, the gas flow in the flow-cell produces a shear stress over the droplet, causing the beads to move. The movements were monitored using an optical microscope and recorded every 1 s using a CCD camera (Hamamatsu, C11440-42U30, Japan). The bead speeds were converted to viscosity using a calibration curve (Fig. S1), which was derived from bead-speeds at different RH versus the known viscosity of the sucrose/$H_2O$ particles at that RH (Fig. S2). Once the movements became too slow, the poke-and-flow technique was adopted.

110 The poke-and-flow technique was used in viscosity ranges of > ~$10^3$ Pa·s. This technique has been widely used to determine the phase and viscosity of aerosol particles (Murray et al., 2012; Renbaum-Wolff et al., 2013a; Li et al., 2020; Song et al., 2021). In brief, the particles, which were deposited on a hydrophobic surface and conditioned in a flow-cell at target-RH as described above, were poked from their top to bottom using a sharp sterile needle (Jung Rim Medical Industrial, South Korea), controlled by a micro-manipulator (Narishige, model MO-152, Japan). The geometrical changes of the particles before, during

115 and, after poking were observed using an optical microscope (Olympus CKX53 with 40× objective, Japan) and recorded using a CCD camera (Hamamatsu, C11440-42U30, Japan).

For particles of sucrose/$H_2O$ and sucrose/AS/$H_2O$ mixtures with an OIR of 4:1, a half-torus geometry with an inner hole was observed after poking them at target-RH upon dehydration, with the hole then gradually closing to minimize the surface energy. In this case, we measured the experimental flow time ($\tau_{(exp, flow)}$), which is the time for the area of the inner hole to reduce to

120 one-fourth of the original hole area, right after poking. Figure S3 shows an example where the $\tau_{(exp, flow)}$ of sucrose/$H_2O$ and sucrose/AS/$H_2O$ particles for an OIR of 4:1 were measured as 149 s at ~42% RH and 425 s at 28% RH. The $\tau_{(exp, flow)}$ of each particle was then converted to the lower limit of the viscosity using the equation proposed by Sellier et al. (2015). Details of the process and the measured $\tau_{(exp, flow)}$ values are described in Sect. S2 and Fig. S4 of the supplement, respectively. For particles of AS/$H_2O$, and sucrose/AS/$H_2O$ for OIRs of 1:1 and 1:4, we could not measure the $\tau_{(exp, flow)}$ using the poke-and-flow technique

125 because the droplets were supersaturated with respect to AS upon dehydration.

All particles containing sucrose cracked upon poking at a certain RH, and no detectable material flow was monitored for more than 3 h. Figure 1 presents the optical images of particles containing an organic compound of sucrose that show no inflow or outflow for 3 h after poking. In this case, we determined the lower limit of viscosity as ~1×$10^8$ Pa·s based on previous studies





of organic particles (Renbaum-Wolff et al., 2013a; Grayson et al., 2015; Song et al., 2019), though the upper limit to viscosity could not be determined in this study from the poke-and-flow experiments.

**2.3 Optical observation of particles during dehydration**

Particle morphologies were observed optically during dehydration. The detailed procedure and method have been described in previous studies (Ciobanu et al., 2009; Bertram et al., 2011; Ham et al., 2019). In this study, particles were deposited onto a hydrophobic glass slide (HR3-239, Hampton Research, USA), and then the glass slide was placed into the flow-cell coupled to an optical microscope (Olympus BX43, 40x objective, Japan). During the experiments, RH was decreased from ~96% to ~0% RH at a rate of ~0.5% RH min$^{-1}$ at $290 \pm 1$ K. RH calibration was carried out in the same way as described in Sect. 2.2, and the RH uncertainty in the cell was $\pm 1.5\%$. The optical images of particles during experiments were recorded every 10 s.

**2.4 Thermodynamic calculations**

The AIOMFAC model (Zuend et al., 2008, 2011) is a widely used tool for studying aerosol mixing thermodynamics. Equilibrium models based on AIOMFAC can be applied to predict the composition and phase state of aerosol systems (e.g., Zuend and Seinfeld, 2012; Hodas et al., 2015; Bouzidi et al., 2020). This model has been previously used to study mixtures of AS and α-pinene-derived SOA (Zuend and Seinfeld, 2012). The recently integrated viscosity module, AIOMFAC-VISC, can predict the viscosity of aqueous organic mixtures (Gervasi et al., 2020), as well as aqueous electrolyte solutions and aqueous organic–inorganic mixtures (Lilek and Zuend, 2021) using a thermodynamics-based approach, which implies the assumption of a Newtonian fluid behaviour.

The AIOMFAC-VISC module consists of separate treatments for mixtures of water with (neutral) organic components and mixtures of water with various inorganic ions. For organic compounds of potentially unknown viscosity, the pure-component viscosities at a given temperature can be estimated based on the glass transition temperatures, $T_g$, of the components, for example by employing the parameterization introduced by DeRieux et al. (2018) for $T_g$. This is the default approach in AIOMFAC-VISC. Alternatively, when the (measured) pure-component viscosity or the $T_g$ value is known, it can be used in the model calculation. The temperature-dependent viscosity of pure water is computed based on the parameterization by Dehaoui et al. (2015). With AIOMFAC-VISC, the viscosity of an aqueous organic solution is then predicted via a combinatorial-activity-weighted mixing rule and a residual correction term, as detailed by Gervasi et al. (2020). As was done previously in the study by Song et al. (2021) to obtain the best possible model fit for mixtures containing sucrose, the version of AIOMFAC-VISC used in this work includes a special treatment of $H_2O$–ether-group interactions in aqueous sucrose solutions.

For mixtures of water with one or more electrolytes, AIOMFAC-VISC uses an equation based on Eyring's absolute rate theory (Glasstone et al., 1941) to calculate viscosity as a function of the Gibbs energy of activation for the volume per mole of mixture and viscous flow. The AIOMFAC-VISC module treating viscosity calculations for electrolyte solutions contains a set of adjustable parameters that have been fitted primarily to bulk viscosity measurements, which do not always agree with droplet-





based measurements in the composition range where such measurements overlap (Lilek and Zuend, 2021). Two unique fit parameters are included for each ion species, and one unique parameter is included for each potential cation–anion pair in the mixture. In the case of aqueous AS, the model includes two single-ion parameters for $NH_4^+$, two parameters for $SO_4^{2-}$, and one representing specific interaction effects between $NH_4^+$ and $SO_4^{2-}$.

AIOMFAC-VISC includes two mixing models for aqueous organic–inorganic mixtures, enabling the coupling of the distinct approaches used for aqueous organic solutions and aqueous electrolyte solutions. The first is called aquelec mixing, wherein all the water contained in the organic-inorganic mixture is assumed to associate with the ions to form an organic-free aqueous electrolyte solution (only used for the purpose of viscosity calculations). This organic-free subsystem effectively replaces pure water in the full system, and the aqueous organic model from Gervasi et al. (2020) is used to calculate the final mixture

viscosity. In aquelec, interactions between ions and organics are not included explicitly, although they are partly captured by the ion activity coefficients which are used to calculate the Gibbs energy of activation for viscous flow of the organic-free subsystem. The second mixing model is a ZSR style approach, wherein the total aerosol water is artificially partitioned between an electrolyte-free aqueous organic subsystem, and an organic-free aqueous electrolyte subsystem, which is designed in such a way that dry OIR is preserved. In this mixing model, direct interactions between ions and organics are ignored in the viscosity

calculation (indirectly there is an effect since all components affect the water activity of the fully mixed solution, which is matched by the water activities of the subsystems). This has been shown to work reasonably well for non-reactive/non-interacting mixtures, but does not work when ions and organic components are interacting strongly to form aggregates of a particular stoichiometry, for example in the gel-forming mixtures described by Richards et al. (2020a,b). Lilek and Zuend (2021) describe further details about the AIOMFAC-VISC method and modeling options for the viscosity of organic–inorganic

mixtures.

## 3 Results and Discussion

### 3.1 Viscosities in binary systems of sucrose/H₂O and AS/H₂O

The viscosity of sucrose/H₂O particles has already been determined by many studies using different techniques at a temperature of 293 - 297 K (Hony et al., 2013; Power et al., 2013; Song et al., 2015; Song et al., 2016b; Song et al., 2021). In this study,

the viscosity of sucrose/H₂O particles gradually increased from $\sim 6 \times 10^{-1}$ to $> \sim 1 \times 10^8$ Pa·s as RH decreased from ~83% to ~24% at a temperature of 293 ± 1 K (Fig. 2a). Our result agrees with the results of previous works as shown in Fig. 2a. From the values of viscosity, sucrose/H₂O particles were determined to be in a liquid state for RH greater than ~69%, a semisolid state for RH values between ~36% and ~69%, and semisolid or even a solid state for RH values less than ~24%.

Figure 2b also illustrates the RH-dependent viscosities of AS/H₂O particles. The mean viscosities at 293 ± 1 K of AS/H₂O

particles determined by the bead-mobility experiment increased from $\sim 6 \times 10^{-3}$ to $\sim 1 \times 10^{-2}$ Pa·s for mean RH ranging from ~61% to ~50% (at ~50% RH, we only observed bead-movements with non-effloresced particles). This corresponds to a liquid state for RH > ~50%.



Upon dehydration, AS/H$_2$O particles effloresced in the RH range between ~50 and ~40% (Fig. 3a), which is a well-known ERH range of pure AS (Winston and Bates, 1960). At ~50% RH, effloresced particles and non-effloresced particles coexisted on the substrate, and when we poked the AS/H$_2$O particles, all particles including non-effloresced particles cracked (Fig. 1b). The act of poking the particles at RH close to that ERH level may induce efflorescence of the particles at that point. This is given that the needle poking action may generate a disturbance that leads to nucleation of a crystal at that point. One likely reason for the observed sharp increase in viscosity is the occurrence of a phase transition from a supersaturated salt solution to a crystalline salt particle (accompanied by the evaporation of water). Such crystallization events during dehydration could occur at RH values slightly higher than the typical range measured for the ERH of the salt in undisturbed droplets, because the action of poking the droplet with a needle for the viscosity measurement introduced as disturbance. That disturbance may induce the nucleation (and subsequent growth) of the salt crystal, similar to the well-known process of contact freezing of supercooled cloud droplets (e.g., Ciobanu et al., 2010; Ladino et al., 2011; Hoose and Möhler, 2012). Based on the observed behaviour, the AS/H$_2$O particles were determined to be in a solid state for RH ≤ ~50%.

## 3. 2 Viscosities in ternary systems of sucrose/AS/H$_2$O with different mixing ratios

To mimic atmospherically relevant aerosol particles containing both organics, water and inorganic ions, we mixed sucrose and AS. Moreover, to explore how viscosity varies depending on the mixing ratio of sucrose/AS/H$_2$O particles, we investigated three different OIRs of 4:1, 1:1, and 1:4.

For organic-rich particles of sucrose/AS/H$_2$O (OIR = 4:1), the mean viscosity was quantified from the bead-mobility technique as ~$4 \times 10^{-2}$ to ~$1 \times 10^{1}$ Pa·s as RH decreased from ~80% to ~52% (blue symbols in Fig. 4). From the poke-and-flow technique, the lower limit to the viscosities was determined to be ~$4 \times 10^{3}$ Pa·s for ~35% RH, ~$8 \times 10^{4}$ Pa·s for ~30% RH and ~$1 \times 10^{5}$ Pa·s for ~28% (Fig. 4). At RH of ~ 25%, it was difficult to determine the viscosity using the poke-and-flow technique because the inner hole formed on poking did not close even during 24 h. The particles clearly cracked at ~18% RH as shown in Fig. 1c, and this produces the lower limit to the viscosity of ~$1 \times 10^{8}$ Pa·s (Renbaum-Wolff et al., 2013a; Grayson et al., 2015; Song et al., 2019; Song et al., 2021). A gradual increase in the viscosities of was observed in the particles of sucrose/AS/H$_2$O for an OIR of 4:1, as RH decreased (Fig. 4). Such a gradual increase in viscosity of the organic-rich particles is similar to that of sucrose/H$_2$O particles (Fig. 2a), but the viscosity values of the sucrose/AS/H$_2$O particles at this OIR are approximately two orders of magnitude lower than those of sucrose/H$_2$O particles at the same RH. In addition, the finding of gradual enhancements in the organic-rich particle is in consistent with the result of other recent viscosity studies of SOA particles (Song et al., 2015; Grayson et al., 2016; Hosny et al., 2016; Song et al., 2016a; Song et al., 2019; Gervasi et al., 2020; Maclean et al., 2021; Smith et al., 2021). The viscosities of the sucrose/AS/H$_2$O for OIR = 4:1 corresponds to liquid for RH > ~52%, semisolid for ~18% < RH < ~50%, and semisolid or solid for RH < ~18%. It is interesting to note that any likely crystallization/efflorescence process of AS in the organic-rich particle was not observable in the optical images on the dehydration process down to ~0% RH (Fig. 3b). Previous studies have shown such behaviour that organic-rich or SOA particles also cracked upon poking, without the appearance of crystallization (Song et al., 2019; Song et al., 2021; Smith et al., 2021).





In particles consisting sucrose/AS/H$_2$O particles with an OIR = 1:1, the mean viscosity varied from ~1 × 10$^{-2}$ to ~4 × 10$^1$ Pa·s from ~70% to ~34% RH (yellow symbols in Fig. 4). At ~30% RH, we could not determine the viscosity using the poke-and-flow technique because the droplets were supersaturated with respect to AS upon dehydration. At ~27% RH, the particles cracked when being poked (Fig. 1d), so the lower limit of the viscosity of the particle was estimated as ~1 × 10$^8$ Pa·s. From

the RH-dependent viscosities, sucrose/AS/H$_2$O particles with an OIR = 1:1 existed as liquid for RH > ~34%, semisolid for ~34 % < RH < ~27%, and semisolid or solid for RH < ~27%. As similar of the organic-rich particle (Fig. 3b), no crystallization of the particle with an OIR = 1:1 was optically observed down to ~0% RH (Fig. 3c).

For inorganic-rich particles of sucrose/AS/H$_2$O (OIR = 1:4), the mean viscosity ranged from ~1 × 10$^{-2}$ to ~1 × 10$^{-1}$ Pa·s for an RH range of ~63% - ~43% determined from the bead-mobility technique (red symbols in Fig. 4). This viscosity result of

sucrose/AS/H$_2$O for OIR = 1:4 observed in the RH range (Fig. 4) is similar to the result in the binary system of AS/H$_2$O free of organic particles at the same RH within experimental errors (Fig. 2b). In the RH range from ~40 to ~30%, we could not approach with the poke-and-flow technique to quantify the viscosities of the particles. When we poked the particles, the particles were quickly crushed or stuck to the needle due to the supersaturation of AS. This behaviour was also observed in the particles for OIR = 1:1, but more distinct behaviour with a wide range of RH was presented as inorganic fraction increased.

Upon dehydration, the particles crystallized or effloresced at ~40 ± 2.0% RH (Fig. 3d), which is close to the ERH of an organic-free AS droplet (Winston and Bates, 1960). At lower RH of ~25%, the inorganic-rich sucrose/AS/H$_2$O particles shattered without any flow of the material for 3 h (Fig. 1e), corresponding to the lower limit of the viscosity of ~1 × 10$^8$ Pa·s. These results lead to a drastic enhancement in viscosity of approximately 8 orders of magnitude in RH from 43% to 25% of the particles as illustrated in Fig. 4 (red symbols), which is comparable to the gradual enhancement in viscosity of the organic-rich

particles (blue symbols). Also, recent studies of Richards et al. (2020b) and Song et al. (2021) showed such a drastic increase in viscosities when inorganic species of CaCl$_2$, MgSO$_4$, Ca(NO$_3$)$_2$, or Mg(NO$_3$)$_2$ were mixed with organic compounds (e.g., glucose, or sucrose). Based on the viscosity values, particles with an OIR of 1:4 were in a liquid state for RH > ~43%, in a semi-solid state for ~25% < RH < ~43%, and in a semisolid or solid state for RH < ~25%.

### 3. 3 Comparison of viscosity predictions and measurements

AIOMFAC-VISC predictions for binary sucrose/H$_2$O, Ca(NO$_3$)$_2$/H$_2$O, Mg(NO$_3$)$_2$/H$_2$O, and AS/H$_2$O are shown in Fig. 2. The predicted viscosity for binary sucrose/H$_2$O agrees well within the error of the measurements, but it does not capture the full spread of measurements for RH between 65% and 85%. The binary Ca(NO$_3$)$_2$/H$_2$O, Mg(NO$_3$)$_2$/H$_2$O, and AS/H$_2$O predictions agree well within the error of the measurements for RH < ~25% (and excluding the measurements at 10$^8$ Pa·s that indicate crystallization). Also, the predicted viscosity for Ca(NO$_3$)$_2$/H$_2$O is less than the measured viscosity at RH = ~15%. Unlike the

predicted viscosities for Ca(NO$_3$)$_2$/H$_2$O and Mg(NO$_3$)$_2$/H$_2$O, the AS/H$_2$O prediction curve does not undergo a steep increase for RH < ~10%, due to the weaker influence of cation–anion pairs for this system compared to systems containing a divalent cation (e.g., Ca$^{2+}$ or Mg$^{2+}$). We also note that the AIOMFAC-VISC calculations were carried out for single liquid phases of



these salt solutions, in which crystallization is suppressed, providing an estimate of the liquid-state viscosity for conditions where the salt would crystallize in an experiment.

Shown in Fig. 4 are the measurement and model comparisons of the RH-dependent viscosities of the sucrose/AS/$H_2O$ systems with OIRs of 4:1, 1:1, and 1:4. The model predictions and measurements agree well for all three OIRs, with the model predicting viscosities mostly within the error of the measurements. For the system studied in this work, the predictions using the ZSR mixing approach for viscosity (Fig. 4) are consistently better than those using the aquelec mixing approach (Fig. S5). The ZSR mixing approach likely performs better than aquelec because it puts more weight on the binary aqueous sucrose

contribution to the predicted mixed-phase viscosity. By contrast, aquelec puts more weight on the viscosity contribution from the aqueous electrolyte solution (i.e., AS/$H_2O$), driving the organic–inorganic mixture viscosity lower. This is especially noticeable for the OIR = 1:4 curve in Fig. S5 when compared to that in Fig. 4.

    The most significant deviations between the model and the measurements are for OIR = 1:1 and OIR = 1:4 at RH below 30%, and these are expected based on similar findings in the study by Song et al. (2021). When particles cracked during poke-and-

flow experiments, perhaps due to crystallization of the solutes, the lower limit of viscosity was reported as ~$10^8$ Pa·s. In these calculations with AIOMFAC-VISC, the predictions do not account for phase changes such as crystallization (or a gel transition), which explains why the model does not reproduce the measurements at these points. However, the model provides an estimate for the viscosity of the particle phase had AS not crystallized. For the organic-rich case (OIR = 4:1), it is expected that the relative abundance of sucrose impedes or completely inhibits crystallization, such that a smooth increase in viscosity

is observed instead, with very good model–measurement agreement.

## 4 Summary

In this study, the viscosities of particles in binary systems of sucrose/$H_2O$ and AS/$H_2O$, and ternary systems of sucrose/AS/$H_2O$ for OIRs of 4:1, 1:1, and 1:4 for decreasing RH, were quantified by bead-mobility and poke-and-flow techniques at 293 ± 1 K. Viscosity of sucrose/$H_2O$ particles increased gradually from ~$6 \times 10^{-1}$ to > ~$1 \times 10^8$ Pa·s when the RH decreased from ~83%

to ~24% while the viscosity of AS/$H_2O$ particles increased drastically to ~$10^{12}$ Pa·s at ~50% RH. For the ternary systems, the viscosity of organic-rich (OIR = 4:1) particles gradually increased from ~$4 \times 10^{-2}$ to ~$1 \times 10^8$ Pa·s with a reduction of RH from ~80% to ~18%, showing a tendency similar to that of the sucrose/$H_2O$ particle; however, viscosities observed were ~2 orders of magnitude lower in the sucrose/AS/$H_2O$ particles. Particles consisting of sucrose/AS/$H_2O$ with an OIR of 1:1 ranged in viscosity from ~$1 \times 10^{-2}$ to ~$1 \times 10^8$ Pa·s for an RH range of ~70 - ~27%. When the fraction of the inorganic salt was higher

(OIR = 1:4), sharp enhancement of viscosity was indicated with values of ~$1 \times 10^{-2}$ to ~$1 \times 10^8$ Pa·s for RH ranging from ~63% to ~25% as in the case of AS/$H_2O$ particles. Based on the viscosity results, the particles of binary and ternary systems ranged from liquid to semisolid, and even the solid state depending on the RH. Moreover, the values of measured viscosities of sucrose/AS/$H_2O$ particles were compared to the values of AIOMFAC-VISC predicted viscosities of sucrose/AS/$H_2O$ particles for OIRs of 4:1, 1:1, and 1:4. The model–measurement comparison showed very good agreement considering the uncertainties



and sensitivity estimates of the measurements and the model, at least for the RH range in which crystallization of the salt could be ruled out. Comparison of the predictions for the three OIRs also suggests that the ZSR-based viscosity prediction approach for organic–inorganic mixtures is superior to that of the "aquelec" method for the sucrose/AS/$H_2O$ system.

*Data availability*. Underlying material and related data for this paper are provided in the Supplement.


*Author contributions*. MS designed this study. RJ and MS conducted viscosity experiments and analysed the data. JL and AZ conducted AIOMFAC-VISC model predictions and wrote the related sections. RJ and MS prepared the article with contributions from JL, AZ, RX, MNC, DK, and HM.

*Competing interests*. The authors declare that they have no conflict of interest.

*Acknowledgements*. Mijung Song gives thanks to Hyeok Jin Kim for the technical support.

*Financial support*. This work was supported by the National Research Foundation of Korea (NRF) grant funded by the Korea
government (MSIT) (NRF-2019R1A2C1086187), by the Fine Particle Research Initiative in East Asia Considering National Differences (FRIEND) Project (NRF-2020M3G1A1114548), and by the Technology Development Program to Solve Climate Changes of the National Research Foundation (NRF) funded by the Korea government (MSIT) (NRF-2019M1A2A2103956). This project was undertaken with the financial support of the government of Canada through the federal Department of Environment and Climate Change (grant no. GCXE20S049). This work was also supported by Alfred P. Sloan Foundation
under Prime Award no. G-2020-13912.

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





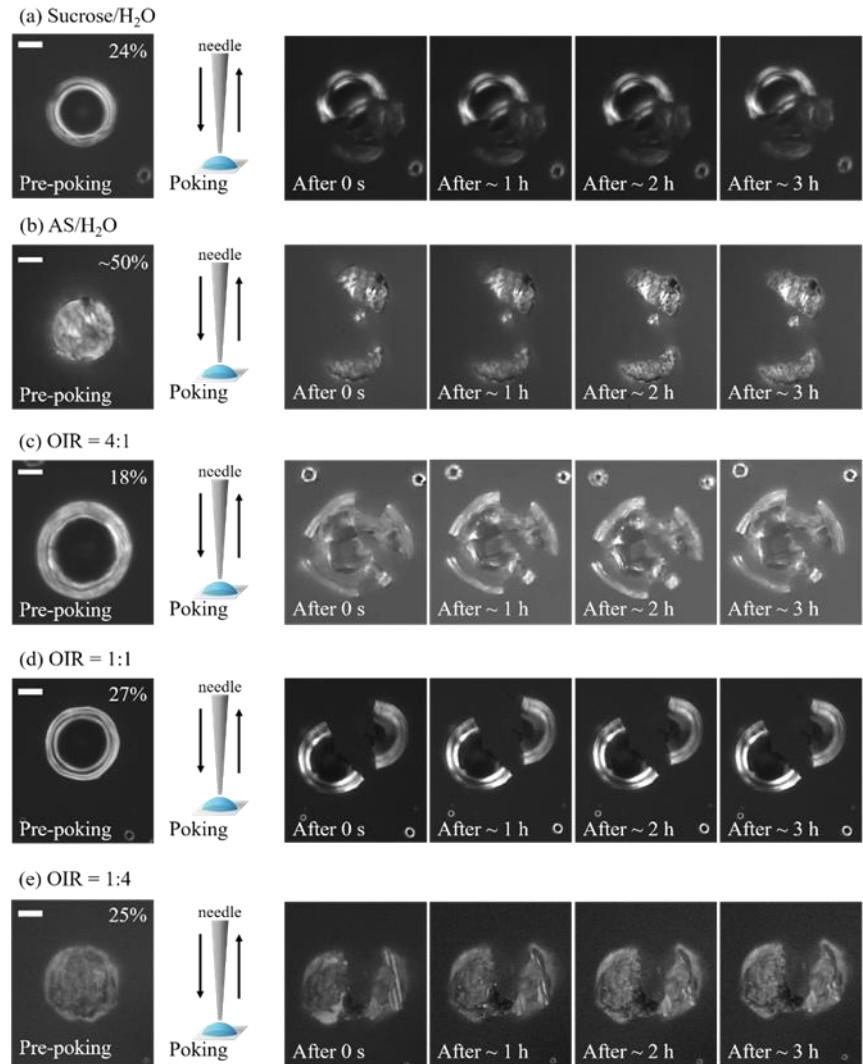

**Figure 1.** Optical images of particles during poke-and-flow experiments at 293 ± 1 K. When particles cracked at certain relative humidity (RH), they were observed for 3 h and no restorative flow occurred. During pre-poking, poking, and post-poking process, the RH was maintained. Images are sequentially presented for (a) sucrose/$H_2O$, (b) ammonium sulfate (AS)/$H_2O$, (c) sucrose/AS/$H_2O$ particles for organic-to-inorganic dry ratios (OIRs) = 4:1, (d) sucrose/AS/$H_2O$ particles for OIR = 1:1, and (e) sucrose/AS/$H_2O$ particles for OIR = 1:4. White scale bar indicates 10 μm.

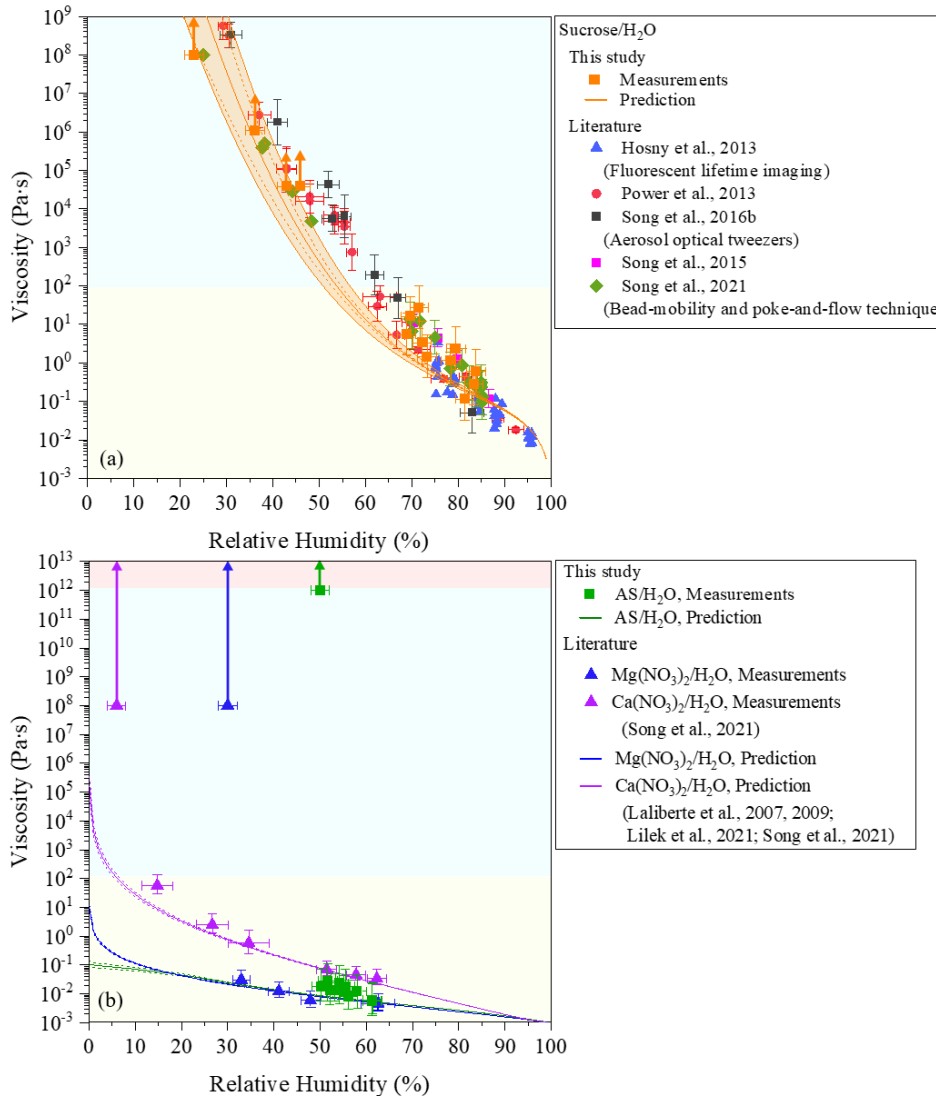

**Figure 2.** Viscosities of measurements AIOMFAC-VISC predictions of binary systems consisting of (a) sucrose/$H_2O$ and (b) ammonium sulfate (AS)/$H_2O$ at 293 ± 1 K. Mean viscosities are results from bead-mobility experiment with error of $x$-axis representing standardization of 3 - 5 beads in one or two particles at given relative humidity (RH). Error of the $y$-axis is produced by 95% prediction bands of calibration curve for bead speed-to-viscosity (Fig. S1). Arrows indicate lower limits to viscosity obtained from poke-and-flow experiment (Sect. S2) determined from experimental flow time (Fig. S4) and the equation by Sellier et al. (2015). When particles containing sucrose cracked due to poking, a lower limit of the viscosity of the particle is determined to be ~$1 \times 10^8$ Pa·s. Light yellow region: liquid phase, light blue region: semi-solid phase, and light red region: solid phase. The viscosity of sucrose/$H_2O$ particles agrees with the results of previous studies included in (a). Previous studies on viscosities of measurement and AIOMFAC-VISC predictions for binary systems of $Mg(NO_3)_2$/$H_2O$, and $Ca(NO_3)_2$/$H_2O$ from Song et al. (2021) are also included in (b). AIOMFAC-VISC predictions with the Zdanovskii–Stokes–Robinson (ZSR)-style organic–inorganic mixing model for viscosity. Model sensitivity, defined by the impact of ± 2% variation in aerosol water content, is shown by the dashed curves. The orange shaded region in (a) shows the potential viscosity prediction error introduced by ± 5% error in the glass transition temperature of sucrose.





**Figure 3.** Optical images on dehydration process at a temperature of $290 \pm 1$ K for (a) ammonium sulfate (AS)/$H_2O$, (b) sucrose/AS/$H_2O$ for organic-to-inorganic dry ratios (OIRs) = 4:1, (c) sucrose/AS/$H_2O$ particles for OIR = 1:1 and (d) sucrose/AS/$H_2O$ particles for OIR = 1:4 at a rate of dehydration of 0.3 - 0.5% RH min$^{-1}$. White scale bar indicates 20 μm.





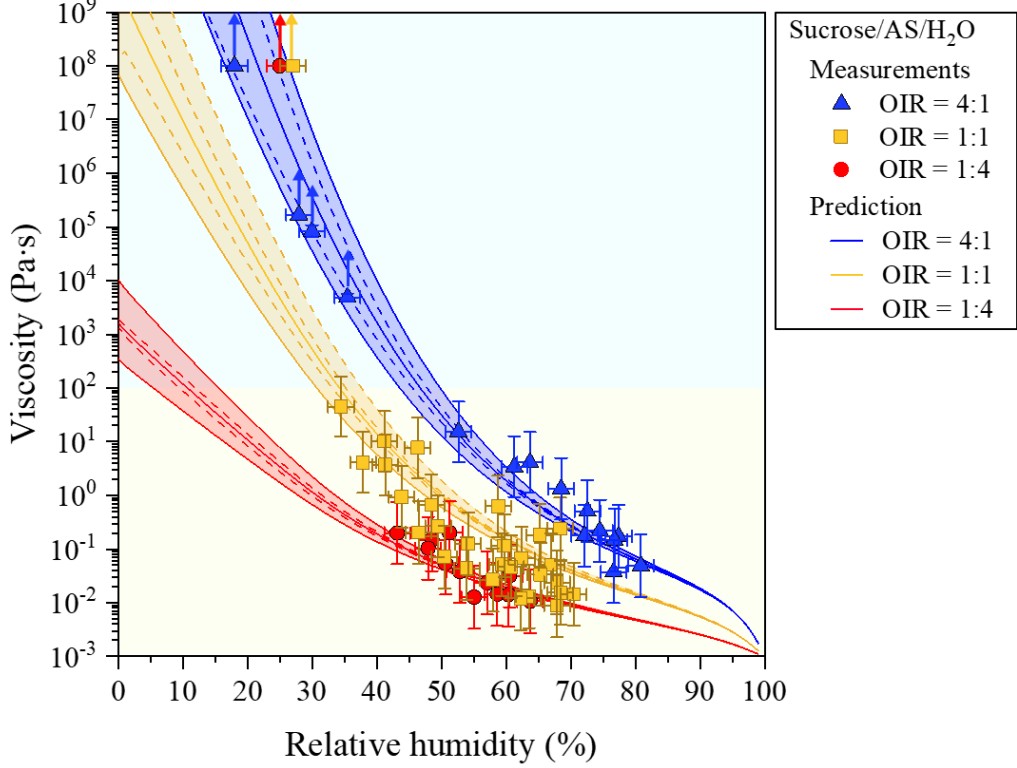

**Figure 4**. Viscosities as a function of relative humidity (RH), with measurements from bead-mobility and poke-and-flow experiments and AIOMFAC-VISC predictions of ternary systems of sucrose/ammonium sulfate (AS)/H$_2$O particles for organic-to-inorganic dry ratios (OIRs) = 4:1, 1:1, and 1:4 at 293 ± 1 K. Arrows indicate lower limits to viscosity obtained from poke-and-flow experiment (Sect. S2) determined from the experimental flow time (Fig. S4) and equation reported from Sellier et al. (2015). When particles containing sucrose cracked by poking, the estimated lower limit of the viscosity of a particle was ~1 × 10$^8$ Pa·s. Mean viscosities shown are the result of bead-mobility experiment with the error along the *x*-axis direction representing standardization of 3 - 5 beads in one or two particles at given RH. The error along the *y*-axis is produced by 95% prediction bands of the calibration curve for bead speed-to-viscosity (Fig. S1). Light yellow region: liquid phase, and light blue region: semi-solid phase. AIOMFAC-VISC predictions with the Zdanovskii–Stokes–Robinson (ZSR)-style organic–inorganic mixing model for viscosity. Model sensitivity, defined by the impact of ± 2% variation in aerosol water content, is shown by the dashed curves. Shaded regions show the potential viscosity prediction error introduced by ± 5% error in the glass transition temperature of sucrose. See Fig. S5 for comparison to AIOMFAC-VISC predictions using the aquelec mixing model.