# Peer review of "Viscosity and physical state of sucrose mixed with ammonium sulfate droplets"

_Atmospheric Chemistry and Physics, 2022_

## Author Comment (AC1)

We thank the reviewers for carefully reading our manuscript and for their valuable comments. Listed below are our responses in blue font addressing the general and specific comments from the reviewers of our manuscript.

**Anonymous Referee #1**

Summary: This manuscript reports measurements of the viscosity of internal mixtures of sucrose and ammonium sulfate as a function of relative humidity across a range of organic to inorganic mixing ratios. The results are compared with the predictions from the AIOMFAC-VISC model. Overall this is a nice piece of work applying established methods to quantify viscosity in a previously under-explored chemical system. The observations are clearly presented and the interpretations are well-supported. The manuscript is well-written and the figures are effective in conveying the relevant data.

General Comments of Referee #1

*[1]* One area that could warrant a deeper discussion and further expand the scope of this work is that of the induced efflorescence of supersaturated mixtures that are rich in AS. The assertion is that the needle allows nucleation of a crystal phase. In particles that also contain sucrose (Figure 3d), the solid phase that nucleates appears to be multiphase. Can the authors discuss the phase of these particles? Are they phase-separated (i.e. AS rich domains and sucrose-rich domains), gel particles (e.g. solid with aqueous / viscous fluid in the void space), well-mixed etc.?

*[A1]* Thank you for your comment. This is a good point! As we discussed in Sect. 3.1 (lines: 196 – 204), AS/$H_2O$ particles effloresced in the RH range between ~50 and ~40% upon dehydration, which is a well-known ERH range of pure AS particles. At ~50% RH, a population of effloresced particles and non-effloresced particles coexisted on the substrate and when the needle poked the particles, all particles, including non-effloresced particles, cracked as shown in Fig. 1b. Although the act of poking the particles at a RH close to the ERH of AS may induce nucleation of a crystal, similar to the well-known process of contact freezing of supercooled cloud droplets (e.g., Ciobanu et al., 2010; Ladino et al., 2011; Hoose and Möhler,

2012). Based on the observed behaviour, the AS/$H_2O$ particles were determined to be in a solid state for RH ≤ ~50%. To make it clearer, we have modified the paragraph to the following:

"Upon dehydration, AS/$H_2O$ particles effloresced in the RH range between ~50 and ~40% (Fig. 3a), which is a well-known ERH range of pure AS (Winston and Bates, 1960). At ~50% RH, a population of effloresced particles and non-effloresced particles coexisted on the substrate, and when the needle poked the AS/$H_2O$ particles, all particles including non-effloresced particles cracked (Fig. 1b). The act of poking non-effloresced particles at a RH close to the ERH of AS may induce the nucleation of an AS crystal, similar to the well-known process of contact freezing of supercooled cloud droplets (e.g., Ciobanu et al., 2010; Ladino et al., 2011; Hoose and Möhler, 2012). All particles, regardless of whether already effloresced or not, cracked when poked at a RH ≤ ~50%. Moreover, when we tried to poke the particles at ~55 % RH, the $\tau_{(exp, \, flow)}$ of the particles was fast, corresponding to a liquid-like physical state and flow behaviour.. Based on the observed contrasting behaviour at lower RH, the AS/$H_2O$ particles were determined to be in a solid state for RH ≤ ~50%."

Regarding the sucrose/AS particles for an OIR of 1:4 (AS-rich particles), the optical imaging method does not allow us to conclusively determine the composition of each phase in those multiphase particles. Thus, while the presence of a crystalline AS phase is likely (compare Figs. 3a and 3d), it is unclear whether the remaining liquid forms a more structured gel state or an amorphous viscous semisolid or solid state. This ternary system may therefore be of interest for future studies employing other probing techniques and phase composition analysis. Moreover, the sucrose/AS particles did not show liquid-liquid phase separation upon dehydration because of the high O:C ratio of 0.92. Liquid-liquid phase separation in organic/inorganic aerosol particles occurs generally for O:C < 0.80 which is already well-known (Bertram et al., 2011; Song et al., 2012; You et al., 2014).

References:

Bertram, A., Martin, S., Hanna, S., Smith, M., Bodsworth, A., Chen, Q., Kuwata, M., Liu, A., You, Y., and Zorn, S.: Predicting the relative humidities of liquid-liquid phase separation, efflorescence, and deliquescence of mixed particles of ammonium sulfate, organic material, and water using the organic-to-sulfate mass ratio of the particle and the oxygen-to-carbon elemental ratio of the organic component, Atmos. Chem. Phys., 11, 10995-11006,

https://doi.org/10.5194/acp-11-10995-2011, 2011.

Ciobanu, V. G., Marcolli, C., Krieger, U. K., Zuend, A., and Peter, T.: Efflorescence of ammonium sulfate and coated ammonium sulfate particles: Evidence for surface nucleation, Phys. Chem. A, 114, 9486-9495, https://doi.org/10.1021/jp103541w, 2010.

Hoose, C.and Möhler, O.: Heterogeneous ice nucleation on atmospheric aerosols: a review of results from laboratory experiments, Atmos. Chem. Phys., 12, 9817-9854, https://doi.org/10.5194/acp-12-9817-2012, 2012.

Ladino, L., Stetzer, O., Lüönd, F., Welti, A., and Lohmann, U.: Contact freezing experiments of kaolinite particles with cloud droplets, Geo. Res. Atms, 116, https://doi.org/10.1029/2011JD015727, 2011.

Song, M., Marcolli, C., Krieger, U., Zuend, A., and Peter, T.: Liquid-liquid phase separation and morphology of internally mixed dicarboxylic acids/ammonium sulfate/water particles, Atmos. Chem. Phys., 12, 2691-2712, https://doi.org/10.5194/acp-12-2691-2012, 2012.

You, Y., Smith, M. L., Song, M., Martin, S. T., and Bertram, A. K.: Liquid–liquid phase separation in atmospherically relevant particles consisting of organic species and inorganic salts, Int. Rev. Phys. Chem., 33, 43-77, https://doi.org/10.1080/0144235X.2014.890786, 2014.

Winston, P. W.and Bates, D. H.: Saturated solutions for the control of humidity in biological research, Ecology, 41, 232-237, https://doi.org/10.2307/1931961, 1960.

---

## Author Comment (AC2)

We thank the reviewers for carefully reading our manuscript and for their valuable comments. Listed below are our responses in blue font addressing the general and specific comments from the reviewers of our manuscript.

**Anonymous Referee #2**

Summary: This manuscript details the viscosity measurement of organic-inorganic mixed droplets with varying RH at room temperature and shows better comparison results with AIOMFAC-VISC makes this a solid paper and provides important dataset. This manuscript is very appropriate for ACP and only minor revisions are needed. There are a few points I'd like to ask the authors to consider:

General Comments of Referee #2

*[1]* Starting in the Abstract, the physical state performance of organic-inorganic mixed droplets has not been highlighted as viscosity. It's better to show the main part of physical state from the results. In the Introduction, physical state is mentioned by describing the phase transition between liquid and solid state. Does the phase state equals to physical state? Aerosol particles are frequently internally mixed, but also shows phase separation with different state. The use of physical state needs to be clear in the paper.

*[A1]* Thank you for the comment and suggestion. Perhaps a brief clarification: we distinguish between the terms "viscosity" and "physical state". Strictly, the physical states of relevance here (in the classical sense) are gaseous, liquid, and solid. However, in the context of viscous liquids, additional terms like a "semisolid state" are widespread to characterize different physicochemical or mechanical properties of viscous (liquid) materials. Hence, while viscosity provides a quantifiable way to distinguish among "liquids", we also emphasize in this study the occurrence of phase transitions and associated changes in physical state, e.g. from (viscous) liquid to crystalline solid. The term phase state is typically used synonymous with physical state, but in the context of viscosity of liquids, states like semisolid may be considered a distinct phase state (but not a distinct physical state). For consistency, we use the term physical state and avoid the term phase state in the revised manuscript. To address the reviewer's comment, we have modified several sentences of the Abstract to the following:

"Herein, we quantified viscosities at $293 \pm 1$ K upon dehydration for the binary systems, sucrose/$H_2O$ and ammonium sulfate (AS)/$H_2O$, and the ternary systems, sucrose/AS/$H_2O$ for organic-to-inorganic dry mass ratios (OIRs) = 4:1, 1:1, and 1:4. For binary systems, the viscosity of sucrose/$H_2O$ particles gradually increased from $\sim 4 \times 10^{-1}$ to $> \sim 1 \times 10^8$ Pa·s when the relative humidity (RH) decreased from $\sim 81\%$ to $\sim 24\%$ ranging from liquid to semisolid or solid state, which agrees with previous studies. The viscosity of AS/$H_2O$ particles remained in the liquid state ($< 10^2$ Pa·s) for RH $> \sim 50\%$, while for RH $\leq \sim 50\%$, the particles showed a viscosity of $> \sim 1 \times 10^{12}$ Pa·s, corresponding to a solid state. In case of the ternary systems, the viscosity of organic-rich particles (OIR = 4:1) gradually increased from $\sim 2 \times 10^{-1}$ to $\sim 1 \times 10^8$ Pa·s for a RH decrease from $\sim 81\%$ to $\sim 18\%$, similar to the binary sucrose/$H_2O$ particles. In the ternary particles for OIR = 1:1, the viscosities ranged from less than $\sim 1 \times 10^2$ for RH $> 34\%$ to $> \sim 1 \times 10^8$ Pa·s at $\sim 27\%$ RH. Compared to the organic-rich particles, in the inorganic-rich particles (OIR = 1:4), drastic enhancement in viscosity was observed as RH decreased; the viscosity increased by approximately 8 orders of magnitude during a decrease in RH from 43% to 25%. Based on the collected viscosity data, all particles studied in this work were observed to exist as a liquid, semi-solid or solid depending on the RH."

 [2] P3L68: '…the ozone uptake coefficient of semi-solid particles was approximately one order of magnitude less than that of liquid particles…' Is the one order of magnitude very important and show much impact on the further reaction? This sentence did not highlight the importance of phase transition.

[A2] To address the referee's comment, we have modified this sentence to the following:

"For example, Steimer et al. (2015) showed that the ozone uptake coefficient of semi-solid particles was approximately one order of magnitude less than that of liquid particles. This result can influence significantly the reaction limitation of mass transport."

Reference:

Steimer, S. S., Berkemeier, T., Gilgen, A., Krieger, U. K., Peter, T., Shiraiwa, M., and Ammann, M.: Shikimic acid ozonolysis kinetics of the transition from liquid aqueous solution to highly viscous glass, phys. Chem. Chem. Phys., 17, 31101-31109,

https://doi.org/10.1039/C5CP04544D, 2015.

*[3]* P5L131: Optical observation of particles during dehydration: It should be notice why the optical observation is needed in the viscosity measurement experiment. It seems to provide direct evidence that when the droplets effloresce and the poke and flow test limitation occurs. This should be mentioned in the discussion part.

*[A3]* To address the referee's comment, the following text has been added to Section 2.3 of the revised manuscript.

"To confirm whether the particles studied undergo efflorescence or not during dehydration, particle morphologies were observed optically."

*[4]* P7L215: '…A gradual increase in the viscosities of was observed…' "of" can be removed.

*[A4]* We have now corrected it.

*[5]* Figure 3: Optical images use different absolute length of white scale to indicate 20 µm among 4 subfigures. It seems that the viscosity measurement detect among 20 -100 µm droplets at random. Does the droplet size influence the measurement uncertainty between bead-mobility and poke-and-flow techniques?

*[A5]* We did not observe a size dependence for the relatively narrow range of sizes investigated during the bead-mobility and poke-and-flow experiments. Renbaum-Wolff et al. (2013) and Rovelli et al. (2019) also showed viscosities with no significant difference in the micrometer-sized range of particles at a given relative humidity.

References:

Renbaum-Wolff, L., Grayson, J., and Bertram, A.: New methodology for measuring viscosities in small volumes characteristic of environmental chamber particle samples, Atmos. Chem. Phys., 13, 791-802, https://doi.org/10.5194/acp-13-791-2013, 2013.

Rovelli, G., Song, Y.-C., Maclean, A. M., Topping, D. O., Bertram, A. K., and Reid, J. P.:

Comparison of approaches for measuring and predicting the viscosity of ternary component aerosol particles, Anal. Chem., 91, 5074-5082, https://doi.org/10.1021/acs.analchem.8b05353, 2019.

*[6]* Figure 4: As the author mentioned, the red dots do not cover the ~30 – 40% RH before the cracking RH (~25%) by using the poke and flow technique. Why does the bead mobility method cannot measure the droplets between 30 – 40% RH? It should be the large variation through liquid to semi-solid phase transition, and the bead mobility technique should be able to measure the viscosity up to $10^3$ Pa s. It needs to explain here.

*[A6]* To address the referee's suggestion, we have now added the following text in Sect. 3.2 (lines: 239 – 242).

"In the RH range from ~40 to ~30% we could not quantify the viscosities of the particles with sufficient accuracy, neither with the bead-mobility nor the poke-and-flow techniques. In this RH range, the bead movements inside the particles were too slow to observe and quantify. In addition, when we poked the particles, the particles would stick to the needle, rendering that approach unsuitable. "

*[7]* Figure 4: "…Mean viscosities shown are the result of bead-mobility experiment with the error along the x-axis direction representing standardization of 3 - 5 beads in one or two particles at given RH." "shown" can be removed.

*[A7]* We have corrected it in the revised manuscript.

*[8]* Figure 4: Does the viscosity measurement of sucrose and AS mixed droplets have the literature results to compare. This organic-inorganic mixed system is common and usually been chosen for lab experiment. More comparison of the viscosity data obtained by different techniques are needed.

*[A8]* Thank you for the comment. Right. This sucrose/AS system is common and has been chosen for other laboratory studies; however, studies on viscosity are limited. Very recently, a paper of Tong et al. (2022) showed the viscosity of sucrose/AS droplet for OIR = 1:1 using an optical tweezer setup at 297 K. We have now added their data points in Fig. 4 and rephrased

related sentences (lines: 232 – 235).

"Results showed that viscosities for sucrose/AS droplet from this study and Tong et al. (2022) are consistent within ~1 order of magnitude at given RH. The viscosity deviations at give RH when comparing the two series of measurements may come from uncertainties associated with the different techniques, temperature ranges, and mode of RH changes (i.e. decreasing or increasing RH)."

Reference:

Tong, Y.-K., Liu, Y., Meng, X., Wang, J., Zhao, D., Wu, Z., and Ye, A.: The relative humidity-dependent viscosity of single quasi aerosol particles and possible implications for atmospheric aerosol chemistry, Phys. Chem. Chem. Phys., https://doi.org/10.1039/D2CP00740A, 2022.

---

## Author Response (AR3)

We thank the reviewers for carefully reading our manuscript and for their valuable comments. Listed below are our responses in blue font addressing the general and specific comments from the reviewers of our manuscript.

**Anonymous Referee #1**

Summary: This manuscript reports measurements of the viscosity of internal mixtures of sucrose and ammonium sulfate as a function of relative humidity across a range of organic to inorganic mixing ratios. The results are compared with the predictions from the AIOMFAC-VISC model. Overall this is a nice piece of work applying established methods to quantify viscosity in a previously under-explored chemical system. The observations are clearly presented and the interpretations are well-supported. The manuscript is well-written and the figures are effective in conveying the relevant data.

General Comments of Referee #1

*[1]* One area that could warrant a deeper discussion and further expand the scope of this work is that of the induced efflorescence of supersaturated mixtures that are rich in AS. The assertion is that the needle allows nucleation of a crystal phase. In particles that also contain sucrose (Figure 3d), the solid phase that nucleates appears to be multiphase. Can the authors discuss the phase of these particles? Are they phase-separated (i.e. AS rich domains and sucrose-rich domains), gel particles (e.g. solid with aqueous / viscous fluid in the void space), well-mixed etc.?

*[A1]* Thank you for your comment. This is a good point! As we discussed in Sect. 3.1 (lines: 196 – 204), $AS/H_2O$ particles effloresced in the RH range between ~50 and ~40% upon dehydration, which is a well-known ERH range of pure AS particles. At ~50% RH, a population of effloresced particles and non-effloresced particles coexisted on the substrate and when the needle poked the particles, all particles, including non-effloresced particles, cracked as shown in Fig. 1b. Although the act of poking the particles at a RH close to the ERH of AS may induce nucleation of a crystal, similar to the well-known process of contact freezing of supercooled cloud droplets (e.g., Ciobanu et al., 2010; Ladino et al., 2011; Hoose and Möhler,

2012). Based on the observed behaviour, the AS/$H_2O$ particles were determined to be in a solid state for RH ≤ ~50%. To make it clearer, we have modified the paragraph to the following:

"Upon dehydration, AS/$H_2O$ particles effloresced in the RH range between ~50 and ~40% (Fig. 3a), which is a well-known ERH range of pure AS (Winston and Bates, 1960). At ~50% RH, a population of effloresced particles and non-effloresced particles coexisted on the substrate, and when the needle poked the AS/$H_2O$ particles, all particles including non-effloresced particles cracked (Fig. 1b). The act of poking non-effloresced particles at a RH close to the ERH of AS may induce the nucleation of an AS crystal, similar to the well-known process of contact freezing of supercooled cloud droplets (e.g., Ciobanu et al., 2010; Ladino et al., 2011; Hoose and Möhler, 2012). All particles, regardless of whether already effloresced or not, cracked when poked at a RH ≤ ~50%. Moreover, when we tried to poke the particles at ~55 % RH, the $\tau_{(exp, flow)}$ of the particles was fast, corresponding to a liquid-like physical state and flow behaviour.. Based on the observed contrasting behaviour at lower RH, the AS/$H_2O$ particles were determined to be in a solid state for RH ≤ ~50%."

Regarding the sucrose/AS particles for an OIR of 1:4 (AS-rich particles), the optical imaging method does not allow us to conclusively determine the composition of each phase in those multiphase particles. Thus, while the presence of a crystalline AS phase is likely (compare Figs. 3a and 3d), it is unclear whether the remaining liquid forms a more structured gel state or an amorphous viscous semisolid or solid state. This ternary system may therefore be of interest for future studies employing other probing techniques and phase composition analysis. Moreover, the sucrose/AS particles did not show liquid-liquid phase separation upon dehydration because of the high O:C ratio of 0.92. Liquid-liquid phase separation in organic/inorganic aerosol particles occurs generally for O:C < 0.80 which is already well-known (Bertram et al., 2011; Song et al., 2012; You et al., 2014).

References:

Bertram, A., Martin, S., Hanna, S., Smith, M., Bodsworth, A., Chen, Q., Kuwata, M., Liu, A., You, Y., and Zorn, S.: Predicting the relative humidities of liquid-liquid phase separation, efflorescence, and deliquescence of mixed particles of ammonium sulfate, organic material, and water using the organic-to-sulfate mass ratio of the particle and the oxygen-to-carbon elemental ratio of the organic component, Atmos. Chem. Phys., 11, 10995-11006,

https://doi.org/10.5194/acp-11-10995-2011, 2011.

Ciobanu, V. G., Marcolli, C., Krieger, U. K., Zuend, A., and Peter, T.: Efflorescence of ammonium sulfate and coated ammonium sulfate particles: Evidence for surface nucleation, Phys. Chem. A, 114, 9486-9495, https://doi.org/10.1021/jp103541w, 2010.

Hoose, C.and Möhler, O.: Heterogeneous ice nucleation on atmospheric aerosols: a review of results from laboratory experiments, Atmos. Chem. Phys., 12, 9817-9854, https://doi.org/10.5194/acp-12-9817-2012, 2012.

Ladino, L., Stetzer, O., Lüönd, F., Welti, A., and Lohmann, U.: Contact freezing experiments of kaolinite particles with cloud droplets, Geo. Res. Atms, 116, https://doi.org/10.1029/2011JD015727, 2011.

Song, M., Marcolli, C., Krieger, U., Zuend, A., and Peter, T.: Liquid-liquid phase separation and morphology of internally mixed dicarboxylic acids/ammonium sulfate/water particles, Atmos. Chem. Phys., 12, 2691-2712, https://doi.org/10.5194/acp-12-2691-2012, 2012.

You, Y., Smith, M. L., Song, M., Martin, S. T., and Bertram, A. K.: Liquid–liquid phase separation in atmospherically relevant particles consisting of organic species and inorganic salts, Int. Rev. Phys. Chem., 33, 43-77, https://doi.org/10.1080/0144235X.2014.890786, 2014.

Winston, P. W.and Bates, D. H.: Saturated solutions for the control of humidity in biological research, Ecology, 41, 232-237, https://doi.org/10.2307/1931961, 1960.

We thank the reviewers for carefully reading our manuscript and for their valuable comments. Listed below are our responses in blue font addressing the general and specific comments from the reviewers of our manuscript.

**Anonymous Referee #2**

Summary: This manuscript details the viscosity measurement of organic-inorganic mixed droplets with varying RH at room temperature and shows better comparison results with AIOMFAC-VISC makes this a solid paper and provides important dataset. This manuscript is very appropriate for ACP and only minor revisions are needed. There are a few points I'd like to ask the authors to consider:

General Comments of Referee #2

*[1]* Starting in the Abstract, the physical state performance of organic-inorganic mixed droplets has not been highlighted as viscosity. It's better to show the main part of physical state from the results. In the Introduction, physical state is mentioned by describing the phase transition between liquid and solid state. Does the phase state equals to physical state? Aerosol particles are frequently internally mixed, but also shows phase separation with different state. The use of physical state needs to be clear in the paper.

*[A1]* Thank you for the comment and suggestion. Perhaps a brief clarification: we distinguish between the terms "viscosity" and "physical state". Strictly, the physical states of relevance here (in the classical sense) are gaseous, liquid, and solid. However, in the context of viscous liquids, additional terms like a "semisolid state" are widespread to characterize different physicochemical or mechanical properties of viscous (liquid) materials. Hence, while viscosity provides a quantifiable way to distinguish among "liquids", we also emphasize in this study the occurrence of phase transitions and associated changes in physical state, e.g. from (viscous) liquid to crystalline solid. The term phase state is typically used synonymous with physical state, but in the context of viscosity of liquids, states like semisolid may be considered a distinct phase state (but not a distinct physical state). For consistency, we use the term physical state and avoid the term phase state in the revised manuscript. To address the reviewer's comment, we have modified several sentences of the Abstract to the following:

"Herein, we quantified viscosities at 293 ± 1 K upon dehydration for the binary systems, sucrose/$H_2O$ and ammonium sulfate (AS)/$H_2O$, and the ternary systems, sucrose/AS/$H_2O$ for organic-to-inorganic dry mass ratios (OIRs) = 4:1, 1:1, and 1:4. For binary systems, the viscosity of sucrose/$H_2O$ particles gradually increased from ~$4 \times 10^{-1}$ to > ~$1 \times 10^8$ Pa·s when the relative humidity (RH) decreased from ~81% to ~24% ranging from liquid to semisolid or solid state, which agrees with previous studies. The viscosity of AS/$H_2O$ particles remained in the liquid state (< $10^2$ Pa·s) for RH > ~50%, while for RH ≤ ~50%, the particles showed a viscosity of > ~$1 \times 10^{12}$ Pa·s, corresponding to a solid state. In case of the ternary systems, the viscosity of organic-rich particles (OIR = 4:1) gradually increased from ~$2 \times 10^{-1}$ to ~$1 \times 10^8$ Pa·s for a RH decrease from ~81% to ~18%, similar to the binary sucrose/$H_2O$ particles. In the ternary particles for OIR = 1:1, the viscosities ranged from less than ~$1 \times 10^2$ for RH > 34% to > ~$1 \times 10^8$ Pa·s at ~27% RH. Compared to the organic-rich particles, in the inorganic-rich particles (OIR = 1:4), drastic enhancement in viscosity was observed as RH decreased; the viscosity increased by approximately 8 orders of magnitude during a decrease in RH from 43% to 25%. Based on the collected viscosity data, all particles studied in this work were observed to exist as a liquid, semi-solid or solid depending on the RH."

 *[2]* P3L68: '…the ozone uptake coefficient of semi-solid particles was approximately one order of magnitude less than that of liquid particles…' Is the one order of magnitude very important and show much impact on the further reaction? This sentence did not highlight the importance of phase transition.

*[A2]* To address the referee's comment, we have modified this sentence to the following:

"For example, Steimer et al. (2015) showed that the ozone uptake coefficient of semi-solid particles was approximately one order of magnitude less than that of liquid particles. This result can influence significantly the reaction limitation of mass transport."

Reference:

Steimer, S. S., Berkemeier, T., Gilgen, A., Krieger, U. K., Peter, T., Shiraiwa, M., and Ammann, M.: Shikimic acid ozonolysis kinetics of the transition from liquid aqueous solution to highly viscous glass, phys. Chem. Chem. Phys., 17, 31101-31109,

https://doi.org/10.1039/C5CP04544D, 2015.

*[3]* P5L131: Optical observation of particles during dehydration: It should be notice why the optical observation is needed in the viscosity measurement experiment. It seems to provide direct evidence that when the droplets effloresce and the poke and flow test limitation occurs. This should be mentioned in the discussion part.

*[A3]* To address the referee's comment, the following text has been added to Section 2.3 of the revised manuscript.

"To confirm whether the particles studied undergo efflorescence or not during dehydration, particle morphologies were observed optically."

*[4]* P7L215: '…A gradual increase in the viscosities of was observed…' "of" can be removed.

*[A4]* We have now corrected it.

*[5]* Figure 3: Optical images use different absolute length of white scale to indicate 20 µm among 4 subfigures. It seems that the viscosity measurement detect among 20 -100 µm droplets at random. Does the droplet size influence the measurement uncertainty between bead-mobility and poke-and-flow techniques?

*[A5]* We did not observe a size dependence for the relatively narrow range of sizes investigated during the bead-mobility and poke-and-flow experiments. Renbaum-Wolff et al. (2013) and Rovelli et al. (2019) also showed viscosities with no significant difference in the micrometer-sized range of particles at a given relative humidity.

References:

Renbaum-Wolff, L., Grayson, J., and Bertram, A.: New methodology for measuring viscosities in small volumes characteristic of environmental chamber particle samples, Atmos. Chem. Phys., 13, 791-802, https://doi.org/10.5194/acp-13-791-2013, 2013.

Rovelli, G., Song, Y.-C., Maclean, A. M., Topping, D. O., Bertram, A. K., and Reid, J. P.:

Comparison of approaches for measuring and predicting the viscosity of ternary component aerosol particles, Anal. Chem., 91, 5074-5082, https://doi.org/10.1021/acs.analchem.8b05353, 2019.

*[6]* Figure 4: As the author mentioned, the red dots do not cover the ~30 – 40% RH before the cracking RH (~25%) by using the poke and flow technique. Why does the bead mobility method cannot measure the droplets between 30 – 40% RH? It should be the large variation through liquid to semi-solid phase transition, and the bead mobility technique should be able to measure the viscosity up to $10^3$ Pa s. It needs to explain here.

*[A6]* To address the referee's suggestion, we have now added the following text in Sect. 3.2 (lines: 239 – 242).

"In the RH range from ~40 to ~30% we could not quantify the viscosities of the particles with sufficient accuracy, neither with the bead-mobility nor the poke-and-flow techniques. In this RH range, the bead movements inside the particles were too slow to observe and quantify. In addition, when we poked the particles, the particles would stick to the needle, rendering that approach unsuitable. "

*[7]* Figure 4: "…Mean viscosities shown are the result of bead-mobility experiment with the error along the x-axis direction representing standardization of 3 - 5 beads in one or two particles at given RH." "shown" can be removed.

*[A7]* We have corrected it in the revised manuscript.

*[8]* Figure 4: Does the viscosity measurement of sucrose and AS mixed droplets have the literature results to compare. This organic-inorganic mixed system is common and usually been chosen for lab experiment. More comparison of the viscosity data obtained by different techniques are needed.

*[A8]* Thank you for the comment. Right. This sucrose/AS system is common and has been chosen for other laboratory studies; however, studies on viscosity are limited. Very recently, a paper of Tong et al. (2022) showed the viscosity of sucrose/AS droplet for OIR = 1:1 using an optical tweezer setup at 297 K. We have now added their data points in Fig. 4 and rephrased

related sentences (lines: 232 – 235).

"Results showed that viscosities for sucrose/AS droplet from this study and Tong et al. (2022) are consistent within ~1 order of magnitude at given RH. The viscosity deviations at give RH when comparing the two series of measurements may come from uncertainties associated with the different techniques, temperature ranges, and mode of RH changes (i.e. decreasing or increasing RH)."

Reference:

Tong, Y.-K., Liu, Y., Meng, X., Wang, J., Zhao, D., Wu, Z., and Ye, A.: The relative humidity-dependent viscosity of single quasi aerosol particles and possible implications for atmospheric aerosol chemistry, Phys. Chem. Chem. Phys., https://doi.org/10.1039/D2CP00740A, 2022.

Daniel Knopf
Co-Editor of Atmospheric Chemistry and Physics

Dear Daniel,

We appreciate your carefully reading and giving us valuable comments. Listed below are our responses to your comments in blue. The manuscript has been revised, accordingly (in blue).

Besides the referees' comments, during the revision, we found our small mistakes of 1) optical observation for efflorescence behavior of sucrose/AS particles with an organic-to-inorganic mixing ratio of 1:1 (Fig. 3c), and 2) production of a sucrose calibration line (Fig. S1) (details are below). These errors have been corrected throughout the manuscript and these changes does not impact the findings and conclusions of this work. We apologize the confusion which we did not elaborate the reason of these changes carefully.

Thank you for handling this manuscript.

With kindest regards,
Mijung Song
Professor of Earth and Environmental Sciences
Jeonbuk National University

Regarding Referee #1: This comment refers to the multiphase nature of the sucrose/AS/$H_2O$ particles with OIR = 1:4. I believe, the referee is looking for the response you are giving starting with "Regarding the sucrose…..". The referee would like the acknowledgment in the manuscript that the multiphase nature of the particle could not be resolved. Most of the text could be included in the manuscript to communicate this to the reader.

➔ Thank you for the comment and suggestion. The following text has been added to Section 3.2 (lines: 249-252) of the revised manuscript.

"At the RH, the particle was observed containing multiphase nature from the optical image (Fig. 3d). While the presence of a crystalline AS phase is likely (compare Figs. 3a and 3d), it is unclear whether the remaining liquid forms a more structured gel state or an amorphous viscous semisolid or solid state. This ternary system may therefore be of interest for future studies employing other probing techniques and phase

composition analysis."

You added a new statement beginning on line 231 without elaborating in the author response.

➔ Thank you for the comment. Regarding the new statement on line 232, when we rechecked our optical images during the revision, we found that the particles with an OIR = 1:1 looked like effloresced at about 28% RH as shown in Fig. 3c but it was not accurately observed optically. Most of the particles with an OIR = 1:1 showed such a morphology. Therefore, we added the new statement as: "At the close RH where particles cracked, the particles crystallized or effloresced although it was not accurately observed optically (Fig. 3c)."

This is bit confusing. In previous sentence, you state particles cracked at 27% RH and refer to Fig. 1d. Then in this new sentence, you state particles effloresce at 28% but this was not well observed and refer to Fig. 3c. The connection you are making here is not clear. This is followed by discussion of Tong et al. data. This feels like a jump. Maybe make clear that you refer back to Fig. 4. There is a typo ("give RH"). Please improve this section

➔ To make it better connection and make it clearer, we have modified the paragraph to the following (lines: 229-240):

"In particles consisting of sucrose/AS/$H_2O$ particles with an OIR = 1:1, the mean viscosity varied from ~5 × 10$^{-2}$ to ~1 × 10$^2$ Pa·s from ~70% to ~34% RH (yellow symbols in Fig. 4). At ~30% RH, we could not determine the viscosity using the poke-and-flow technique because the droplets were supersaturated with respect to AS upon dehydration. The particles cracked by poking at ~27% RH (Fig. 1d), so the lower limit of the viscosity of the particle was estimated as ~1 × 10$^8$ Pa·s (Fig. 4). At the close RH where particles cracked, the particles crystallized or effloresced although it was not accurately observed optically (Fig. 3c). The viscosities of sucrose/AS/$H_2O$ particles for an OIR = 1:1 using an optical tweezer are also included in Fig. 4 (Tong et al., 2022). Results showed that viscosities for sucrose/AS droplet from this study and Tong et al. (2022) are consistent within ~1 order of magnitude at given RH. The viscosity deviations at given RH when comparing the two series of measurements may come

from uncertainties associated with the different techniques, temperature ranges, and mode of RH changes (i.e. decreasing or increasing RH). From the RH-dependent viscosities, our result showed that sucrose/AS/$H_2O$ particles with an OIR = 1:1 existed as liquid for RH > ~34%, semisolid for ~34 % < RH < ~27%, and semisolid or solid for RH < ~27% (Fig. 4).”

Regarding Referee #2, first comment: Your definition of physical state is correct. I believe, the confusion is the following: a liquid-liquid phase separated aerosol particle has a "total" particle physical state of liquid. A particle with a solid and liquid phase, what is its physical state? Multiphase. This comes also back to referee #1's comment: the identification of the physical states of the various phases inside the particle. In the case above, the AS is solid, but the remaining solution may be solid, may be not. In either case, if you poke the particle, it will still shatter. Does this mean the particle is overall solid? Can you state that the physical state of this particle is solid? I believe, these are the main issues of both referees. I feel like referee #2 wants a clear definition of these concepts in the introduction.

➔ You are right! We defined the physical state of "total" aerosol particle in this study. To avoid the confusion, we have revised and added the sentences for the physical states of aerosol particles in the introduction as below:

Line: 43-45

"Physical states (i.e. liquid, semi-solid, and solid) of aerosol particles can be determined from their dynamic viscosities; a viscosity of less than $10^2$ Pa·s indicates a liquid state, a viscosity between $10^2$ and $10^{12}$ Pa·s indicates a semi-solid state, and a viscosity of greater than $10^{12}$ Pa·s indicates a solid state (Zobrist et al., 2008; Koop et al., 2011; Kulmala et al., 2011)."

Lines: 83-84

"Next, we determined the physical states (i.e. liquid, semi-solid, and solid) of the particles as a function of RH based on the viscosity-value of the binary and ternary mixtures. In this study, we defined the physical states of the total aerosol particles."

I do not see any changes in revised abstract with regard to the referee's initial comment

(definitions of viscosity/phase/particle physical state). The referee suggests mentioning the particle physical state as well, i.e., which particle systems are entirely liquid or solid or multiphase in nature.

➔ To address the editor's comment, we have revised the abstract as following:

"Herein, we quantified viscosities at $293 \pm 1$ K upon dehydration for the binary systems, sucrose/$H_2O$ and ammonium sulfate (AS)/$H_2O$, and the ternary systems, sucrose/AS/$H_2O$ for organic-to-inorganic dry mass ratios (OIRs) = 4:1, 1:1, and 1:4 using bead-mobility and poke-and-flow techniques. Based on the viscosity-value of the aerosol particles, we defined the physical states of the total aerosol particles studied in this work. For binary systems, the viscosity of sucrose/$H_2O$ particles gradually increased from $\sim 4 \times 10^{-1}$ to $> \sim 1 \times 10^8$ Pa·s when the relative humidity (RH) decreased from ~81% to ~24% ranging from liquid to semisolid or solid state, which agrees with previous studies. The viscosity of AS/$H_2O$ particles remained in the liquid state ($< 10^2$ Pa·s) for RH > ~50%, while for RH $\leq$ ~50%, the particles showed a viscosity of $> \sim 1 \times 10^{12}$ Pa·s, corresponding to a solid state. In case of the ternary systems, the viscosity of organic-rich particles (OIR = 4:1) gradually increased from $\sim 1 \times 10^{-1}$ to $\sim 1 \times 10^8$ Pa·s for a RH decrease from ~81% to ~18%, similar to the binary sucrose/$H_2O$ particles. This indicates that the sucrose/AS/$H_2O$ particles ranges from liquid to semisolid or solid across the RH. In the ternary particles for OIR = 1:1, the viscosities ranged from less than $\sim 1 \times 10^2$ for RH > 34% to $> \sim 1 \times 10^8$ Pa·s at ~27% RH. The viscosities correspond to liquid for RH > ~34%, semisolid for ~34 % < RH < ~27%, and semisolid or solid for RH < ~27%. Compared to the organic-rich particles, in the inorganic-rich particles (OIR = 1:4), drastic enhancement in viscosity was observed as RH decreased; the viscosity increased by approximately 8 orders of magnitude during a decrease in RH from 43% to 25% resulting in liquid to semisolid or solid in the RH range. Overall, all particles studied in this work were observed to exist as a liquid, semi-solid or solid depending on the RH."

Looking at abstract revision, some sentences changed, and experimentally derived parameters changed as well (also in main text) without explanation in the authors' response. Do these parameter changes impact the findings or conclusions of this work? Please elaborate.

➜ During our final check on the revision, we found a mistake on the production of a sucrose calibration line using bead-mobility technique which was used with formal data in our group. Thus, we have replaced the calibration curve of sucrose using corrected data that were produced from Rani Jeong (the first author of this manuscript) from her experiments (corrected Fig. S1 is below). This modification has changed slightly the viscosity-value and corresponding RH obtained from the bead-mobility experiments, but this is more accurate and it is not significantly affect the results.

[Figure]

Figure S1: Calibration curve showing mean bead speeds as function of viscosities of sucrose/$H_2O$ particles at different relative humidity (RH) values. The red curve is produced by a linear fit to the measurements, which yields the equation: *bead speeds = 7.35×10$^{-4}$ × (viscosity, η)$^{-1.09}$*. The pink shaded envelope indicates 95% prediction bands of fitting to the data in this study. The error in mean bead speed (x-axis) is a standardization of 3-5 beads in one or two particles at given RH.